# Design of Intelligent Firefighting and Smart Escape Route Planning System Based on Improved Ant Colony Algorithm

**DOI:** 10.3390/s24196438

**Published:** 2024-10-04

**Authors:** Nan Li, Zhuoyong Shi, Jiahui Jin, Jiahao Feng, Anli Zhang, Meng Xie, Liang Min, Yunfang Zhao, Yuming Lei

**Affiliations:** 1School of Electrical and Information Engineering, Xi’an Jiaotong University City College, Xi’an 710018, China; 2School of Electronics and Information, Northwestern Polytechnical University, Xi’an 710129, China; 3School of Physics and New Energy, Xi’an Jiaotong University City College, Xi’an 710018, China

**Keywords:** UAV, path planning, monitoring systems, LabVIEW

## Abstract

Due to the lack of real-time planning for fire escape routes in large buildings, the current route planning methods fail to adequately consider factors related to the fire situation. This study introduces a real-time fire monitoring and dynamic path planning system based on an improved ant colony algorithm, comprising a hierarchical arrangement of upper and lower computing units. The lower unit employs an array of sensors to collect environmental data in real time, which is subsequently transmitted to an upper-level computer equipped with LabVIEW. Following a comprehensive data analysis, pertinent visualizations are presented. Capitalizing on the acquired fire situational awareness, a propagation model for fire spreading is developed. An enhanced ant colony algorithm is then deployed to calculate and plan escape routes by introducing a fire spread model to enhance the accuracy of escape route planning and incorporating the A* algorithm to improve the convergence speed of the ant colony algorithm. In response to potential anomalies in sensor data under elevated temperature conditions, a correction model for data integrity is proposed. The real-time depiction of escape routes is facilitated through the integration of LabVIEW2018 and MATLAB2023a, ensuring the dependability and safety of the path planning process. Empirical results demonstrate the system’s capability to perform real-time fire surveillance coupled with efficient escape route planning. When benchmarked against the traditional ant colony algorithm, the refined version exhibits expedited convergence, augmented real-time performance, and effectuates an average reduction of 17.1% in the length of the escape trajectory. Such advancements contribute significantly to enhancing evacuation efficiency and minimizing potential casualties.

## 1. Introduction

In the construction of intelligent buildings, large indoor spaces such as shopping malls and office buildings can easily lead to significant property damage and casualties in the event of a fire due to their complex structures and high foot traffic. Therefore, for large indoor spaces, it is crucial to quickly understand the situation during a fire and provide the best escape routes for trapped individuals.

Previous research has extensively explored the detection, monitoring, and evacuation of personnel during indoor fires. However, there is a lack of synchronization between fire alarm monitoring and path planning. While factors affecting escape routes have been analyzed, the malfunction of sensors in high-temperature fire environments, leading to false alarms and missed reports, has not been considered. At the same time, due to the spread of the fire and its impact on the escape routes [1], optimizing the escape route by combining the shortest path with the fire hazard level is a key focus in research on evacuation in complex environments [2].

To this end, this paper proposes a fire monitoring and escape route planning system based on an improved ant colony algorithm (ACO). The system collects real-time indoor fire situation information using sensors, constructs an indoor fire spread model, plans escape routes using the improved ant colony algorithm [3], corrects abnormal node data, and builds a LabVIEW visualization interface to achieve real-time data display, fire alarms, and dynamic escape route planning [4].

The structure of this article is as follows. Section 2 provides a literature review, while Section 3 describes the proposed fire monitoring and escape route planning algorithm and its application in the real world. Section 4 presents the experiments and data analysis. In Section 5, conclusions and future work are presented.

## 2. Literature Review

In the research on path planning for intelligent vehicles, researchers focus on addressing the problem of how to select appropriate algorithms for path planning in complex and unknown environments. To analyze the advantages of current path planning algorithms, intelligent vehicle path planning algorithms are categorized into traditional path planning methods, intelligent path planning methods, and reinforcement learning (RL) path planning methods [5,6].

Traditional path planning methods mainly include studies on artificial potential field methods, the Dynamic Window Approach (DWA), and improvements to the A* algorithm [7]. Gaoh et al. [8] discussed the limitations of the artificial potential field (APF) method in path planning, especially the issues encountered when the target is unreachable, and provided new ideas for improving the APF method. Xiong et al. [9] focused on optimizing and parallelizing the A* algorithm. By improving the search strategy and algorithm structure, they significantly enhanced the efficiency and computation speed of path planning. Zhang et al. [10] utilized an improved heuristic function to guide the A* algorithm, providing an effective path planning solution for autonomous ground vehicles, thereby enhancing the algorithm’s adaptability and robustness in complex environments. Niu et al. [11] proposed an improved A* algorithm to adapt to dynamic environments and the requirements of multi-vehicle collaborative operations. This study not only enhanced the efficiency of path planning but also expanded the algorithm’s applicability in real-world scenarios.

The main intelligent algorithms in the field of path planning include genetic algorithms, ant colony optimization (ACO), particle swarm optimization, probabilistic roadmaps, and rapidly exploring random trees. Luo et al.’s [12] research proposed a new path planning method for multiple unmanned cleaning vehicles, which utilizes the global search characteristics of genetic algorithms and a step-by-step improvement strategy to optimize the cleaning process. Song [13] proposed a dynamic path planning method that integrates fuzzy logic with an improved ant colony algorithm, aiming to solve real-time path planning problems in complex environments while enhancing the accuracy and adaptability of path planning. Jin and his team [14] developed an improved probabilistic roadmap planning method specifically designed for safe drone flights in indoor environments. By optimizing the construction and exploration strategies of the roadmap, they significantly reduced the risk of collisions. Yang et al. [15] developed a new static obstacle avoidance algorithm for autonomous vehicles. By improving the rapidly exploring random tree (RRT) algorithm, they enhanced the safety performance of vehicles in urban environments. Qi et al.’s [16] work on ‘AUV Path Planning Based on Improved Particle Swarm Optimization’ proposes a method for AUV (Autonomous Underwater Vehicle) path planning that integrates an improved particle swarm optimization (PSO) algorithm. This method not only enhances the efficiency of path planning but also improves the algorithm’s adaptability to environmental changes.

Ant colony optimization (ACO) has been widely applied due to its strong global search capability and parallel search features. Considering the complexity of fire environments, scholars tend to combine ACO with other intelligent algorithms to enhance the adaptability of the algorithm.

Cui et al. [17] proposed a multi-strategy adaptive ant colony optimization algorithm (MsAACO) which employs four improvements to mitigate the insufficient convergence and inefficiency issues of ACO. Experimental statistical results show that MsAACO can effectively enhance search efficiency. Amin Hashemi et al. [18] proposed a heuristic-based ACO algorithm, which uses a multi-criteria decision-making (MCDM) procedure applied to set feature-selection tasks to evaluate the performance of the proposed method. Hongguang Wu et al. [19] introduced an ant colony optimization algorithm with destruction and repair strategies (ACO-DR). To verify the performance of the proposed ACO-DR algorithm, Solomon benchmarks and Gehring–Homberger benchmarks were tested, and comparisons were made with state-of-the-art algorithms. Experimental results show that the ACO-DR algorithm is feasible. Qisong Song et al. [20] proposed a new improved PSO-ACO algorithm based on the hybrid algorithm concept to address the issue of low energy dispatch efficiency between stations. Simulation results show that the improved PSO-ACO algorithm can plan shorter routes, take less time, and ensure higher safety for on-site energy dispatching, achieving comprehensive and global optimization of on-site energy dispatching. Ritesh Bhata et al. [21] introduced a breakthrough application of the Bellman–Ford algorithm in optimizing evacuation routes in multi-story school buildings. This research significantly advanced the field of emergency evacuation planning, offering valuable insights to emergency response practitioners, facility managers, and policymakers. Ibrahim Alameri et al. [22] sought to address the critical need for efficient routing in Mobile Ad Hoc Networks (MANETs). Through detailed algorithm analysis, it was demonstrated that the Dijkstra algorithm provides a highly feasible solution for real-time routing problems in metropolitan area networks, especially under dynamic updates. Experimental results show that, in terms of routing optimization, the computational performance of the Dijkstra algorithm is 30% better than the Bellman–Ford algorithm.

Path planning in the field of reinforcement learning mainly includes model-based methods (policy-based and value-based learning) and model-free methods (such as the Q-learning algorithm and the SARSA algorithm). Liu and others [23] proposed a method using Field of View Set (FOVS) to optimize the path planning of assembly robots. This method emphasizes real-time performance and can effectively handle path planning issues in dynamic environments. Racanière [24] explored how to enhance the capabilities of deep reinforcement learning (DRL) agents by incorporating mechanisms of imagination. The study demonstrated the potential of imagination in improving learning efficiency and final performance. Cui et al. [25] proposed an improved value iteration algorithm combined with reinforcement learning. This algorithm optimizes the decision-making process by leveraging environmental feedback, thereby enhancing the efficiency and quality of path planning. Zhang et al. [26] introduced a path search method that combines neural networks with heuristic reinforcement learning. This method effectively addresses path planning problems in complex environments, enhancing the accuracy and speed of the search process. Qijie et al. [27] integrated the rapidly exploring random tree (RRT) and SARSA(λ) reinforcement learning algorithms to address path planning in unknown and complex environments. This approach enhances the adaptability and robustness of path planning. Liu [28] proposed a method for urban traffic route planning using reinforcement learning. This method can adapt to complex traffic conditions, providing an effective route selection strategy for urban traffic. Munoz [29] explored the application of deep Gaussian processes in machine learning. Although not directly related to path planning, this method holds significant value for understanding and processing nonlinear and high-dimensional data. Nair [30] compared the performance of the traditional Dijkstra algorithm with the temporal difference learning algorithm in path planning. The study shows that in certain cases the learning algorithm can provide solutions that are comparable to or even better than the traditional algorithm. Mnih et al. [31] marked a significant breakthrough in deep reinforcement learning in complex tasks, especially in environments such as Atari games. Although primarily focused on gaming, their methods and ideas also provide insights for fields like path planning. Arulkumaran et al. [32] provide a comprehensive overview of deep reinforcement learning, including its core concepts, key algorithms, and application areas. This review is a valuable resource for researchers looking to understand the prospects of DRL in path planning. Hasselt et al. [33] introduced a novel deep reinforcement learning algorithm that addresses the issue of overestimation through Double Q-learning, thereby improving the stability and performance of the learning process. This method is particularly useful for handling uncertainty and complexity in path planning.

## 3. Proposed Work

The overall system block diagram is shown in Figure 1, consisting of an upper computer and a lower computer. The lower computer is composed of a data acquisition module and a microcontroller. The data acquisition module includes a temperature sensor and a smoke concentration sensor, which collect temperature data and CO concentration data in the air. The microcontroller uploads the data to the upper computer. The upper computer realizes data reception, real-time monitoring, abnormal data alarm, and, through joint debugging with MATLAB 2023 simulation software, performs real-time escape route planning to construct a fire monitoring and escape route planning model.

### 3.1. Escape Route Planning Based on an Improved Ant Colony Algorithm

Due to its advantages such as positive feedback and strong robustness, the ant colony algorithm has been widely applied in path optimization problems. However, at the initial stage, blind searches caused by the lack of pheromones slow down the convergence speed. To shorten the search time of the ant colony algorithm and accelerate convergence, this study integrates the A* algorithm with it. First, a fire situation model is established to improve the ant colony’s evaluation function. Next, the A* algorithm is introduced to enhance the pheromone distribution and heuristic function of the ant colony, and the specific principles are as follows.

#### 3.1.1. Traditional Ant Colony Algorithm

The ant colony algorithm is a simulation-based evolutionary algorithm inspired by the foraging behavior of ants, and it has been widely applied to path planning problems. During foraging, each ant releases pheromones on the paths it travels, and ants tend to choose paths with higher pheromone concentrations. This selection mechanism creates a positive feedback loop: the shorter the path, the higher the probability of it being chosen, resulting in more pheromone accumulation on that path, which further enhances its attractiveness. Over time, the ant colony will gradually tend to choose the path with the highest pheromone concentration, which is the optimal path. In the ant colony algorithm, each path represents a feasible solution to be optimized, and the pheromones represent the quality of that solution. By simulating the foraging process of ants, the algorithm continuously updates the pheromone concentrations and guides the ants to choose better paths, ultimately finding the optimal solution [34].
Traditional pheromone concentration

In a real environment, the pheromones released by ants gradually evaporate over time. Assuming a constant evaporation rate, the path with a higher concentration of pheromones will retain more information over the same period. Based on this, the pheromone update formula is given as follows:(1)$$\tau_{ij}(t+1)=\rho \tau_{ij}(t)+\sum_{k=1}^{M}\Delta \tau_{ij}^{k}$$

Formula (1) represents the amount of pheromone remaining on the path between i and j at time t. At the initial moment, the amount of pheromone on each path is set to the same constant. M is the total number of ants in the ant colony. is the pheromone evaporation coefficient (ranging from 0 to 1), and represents the amount of pheromone left on the path between i and j by the k-th ant in this iteration.
(2)$$\Delta \tau_{ij}^{k}=\left\{\begin{array}{l} \frac{Q}{L_{k}},\mathrm{the}\;k\;\mathrm{th}\;\mathrm{ant}\;\mathrm{crosses}\;\mathrm{the}\;\mathrm{line}\;\mathrm{between}\;i\;\mathrm{and}\;j\;\\ 0,else \end{array}\right.$$

In Formula (2), $L_{k}$ represents the path length traveled by the k-th ant in this iteration, and Q is a constant.
2.Selection probability
(3)$$p_{ij}^{k}(t)=\left\{\begin{array}{l} \frac{\tau_{ij}^{\alpha }(t)\eta_{ij}^{\beta }}{\sum_{s\in allowed}\tau_{i_{s}}^{\alpha }(t)\eta_{i_{s}}^{\beta }},s\in \mathrm{allowed}\\ 0,else \end{array}\right.$$

At time t, the probability of ant k (k = 1, 2, 3, …, M) choosing to move from starting point i to endpoint j is shown in Formula 3. Here, “s∈allowed” refers to the set of passable nodes surrounding the current node i. α and β represent the pheromone concentration coefficient and the heuristic function coefficient, respectively. $\eta_{ij}$ is the heuristic factor at time t, and its value is the reciprocal of the distance from i to j at time t, as shown in Formula 4:(4)$$\eta_{ij}=\frac{1}{d_{ij}}$$

#### 3.1.2. Fire Spread Model

This paper utilizes real-time information from each node regarding the fire situation and constructs a model of the fire threat spread using the Kriging interpolation algorithm, generating real-time evolution maps of the fire threat within the building.

The occurrence of a fire is accompanied by the production of large amounts of toxic gases and high temperatures. In actual buildings, smoke detectors and fire alarms are installed, which can indicate whether harmful gases or temperatures exceed safe levels. This helps to plan escape routes for those trying to evacuate, enabling them to avoid these dangerous areas and reduce casualties.

The evolution of a fire primarily consists of four stages: the initial occurrence of the fire, the mid-stage spread and growth, the late-stage full development, and the fire decay stage. This article focuses on analyzing and discussing the fire spread and escape route planning during the initial occurrence, mid-stage spread and growth, and late-stage full development phases. For the classification of danger levels, this article mainly relies on the environmental temperature and CO gas concentration trends, combined with the human tolerance values for temperature and CO concentration during a fire, dividing it into four danger levels. As shown in the table below, the unit of temperature is Celsius, and the unit of carbon monoxide gas concentration is ppm (ppm is a volumetric concentration, 1 ppm = 1 ${\mathrm{cm}}^{3}$/1 $m^{3}$).

Using the Kriging interpolation algorithm, the fire attribute values at various points within a building are calculated. A fire evolution map is created with a time interval of 0.1 s, providing real-time support for escape path planning. The Kriging interpolation algorithm fits the empirical semi-variogram function based on the spatial distance between discrete samples, calculates weight coefficients using unbiased and optimal estimation, and then estimates the value at unknown points using these weight coefficients. Essentially, it estimates attribute values at non-sampled locations through regionalized variables and is considered one of the most unbiased estimation methods. The purpose of the work by Ilies Benikhlef et al. [35] is to analyze and validate the gravity measurement data in the study area, in order to highlight the valid data on gravity anomalies, confirming the effectiveness of the Kriging algorithm developed in this context. The computation process is divided into two parts: the first part quantifies the spatial correlation of the known function using the variogram function; the second part constructs the Kriging equations by searching within the range of domain points, solves them to obtain weight coefficients, and performs a weighted sum to get the attribute values for the interpolation points. Jianchen Di et al. [36] proposed an ordinary Kriging interpolation method to predict ASF values across the entire test area and used this method to forecast the ASF values throughout the region. Cross-validation was employed to verify our predictions. The results confirmed the accuracy and effectiveness of the Kriging interpolation algorithm in predicting ASF values within a specific area. The variogram ultimately reflects the randomness and correlation of the studied variables in space, so the result of each step will affect the accuracy of the final interpolation result. In the study area, the variable Z(x) takes values $Z(x_{i})$ at the sampling points $x_{i}(i=1,2,3,\ldots ,n)$, and the unknown point variable $Z^{*}(x)$ is obtained by the weighted sum of the above n known variable values, as shown in Formula (5).
(5)$$Z^{*}=\sum_{i=1}^{n}\lambda_{i}Z(x_{i})$$

In Formula (5), $\lambda_{i}(i=1,2,3,\ldots ,n)$ represents the required weight coefficient, indicating the contribution of each spatial sample observation to the estimated value. To obtain the weight coefficient, two conditions must be met: (1) the estimation should be unbiased, meaning the mathematical expectation of the bias is 0; (2) it should be the optimal estimation, meaning the sum of the squared differences between the estimated and actual values is minimized, satisfying the Kriging system of equations as shown in Formula (6).
(6)$$\left\{\begin{array}{l} \sum_{i=1}^{n}\sum_{j=1}^{n}\lambda_{j}\gamma (x_{i},x_{j})+\mu =\gamma (x_{i},x)\\ \sum_{i=1}^{n}\lambda_{i}=1 \end{array}\right.$$

In Formula (6), $x_{i}$ and x represent the positions of the i-th sample point and the unknown point, respectively, used to calculate the semi-variogram γ(h); $\lambda_{i}$ is the weight coefficient; and μ is the Lagrange constant. The Kriging equations were solved to obtain the weight coefficients, and Formula (7) was then used to estimate the unknown points.
(7)$$\gamma (h)=\frac{1}{2}E[x)-Z(x+h){]}^{2}$$

#### 3.1.3. Improved Ant Colony Algorithm Based on A* Algorithm (IACO-A*)

A* algorithm

The A* algorithm is a classic heuristic algorithm based on graph search, proposed by Hart in the 1960s [37]. The basic idea is to discretize the state space into a search graph in a deterministic manner and use heuristic information to find the optimal path. The basic formula is shown as Formula (5).
(8)f(n)=g(n)+h(n)

In Formula (5), f(n) represents the total cost from the initial node to the target node; g(n) represents the actual cost from the initial node to the current node; h(n) represents the estimated cost from the current node to the target node, which is the heuristic function. The value of f(n) depends on the value of h(n)—the closer h(n) is to the actual distance, the lower the total cost.

The advantage of the A* algorithm lies in its rapid response to the environment and its direct pathfinding, which is why it is widely used in path planning problems. When performing path searches using the A* algorithm, the search area can be simplified into a set of quantifiable nodes. To find the shortest path, you start at the initial point, examine its adjacent squares, and then expand outward until the target is found.

For the established fire threat spread model, it is necessary to define a comprehensive evaluation function to assess the extended nodes, thereby selecting the best safe escape route. Based on this, the evaluation function is improved as shown in Formula (9). c(n) represents the value of the effective cost of the path.
(9)f(n)=g(n)+h(n)+c(n)

To comprehensively consider avoiding fire hazards during the personnel evacuation process, the actual cost function g(n) in the above evaluation is defined as Formula (10):(10)$$g(n)=\lambda_{1}L_{n}+\lambda_{2}F_{n}=\lambda_{1}\sqrt{{\left(x_{n}-x_{i}\right)}^{2}+{\left(y_{n}-y_{i}\right)}^{2}}+\lambda_{2}F$$

In Formula (10), $L_{n}$ represents the distance from the current node n to the target node i; $F_{n}$ denotes the fire hazard coefficient at the current node; $x_{i}$ and $y_{i}$ represents the x and y coordinates of the target node; $\lambda_{j}(j=1,2)$ indicates the corresponding indicator weight coefficient, which should be set according to actual requirements.

The evaluation function h(n) is determined by the Euclidean distance, as defined in Formula (11).
(11)$$h(n)=\omega \sqrt{{\left(x_{n}-x_{s}\right)}^{2}+{\left(y_{n}-y_{s}\right)}^{2}}$$

In Formula (11), $x_{s}$ and $y_{s}$ represent the horizontal and vertical coordinates of the final position; ω represents the transfer weight coefficient, which can be adjusted to change the effect of the estimated cost on the path planning.

The definition of c(n) is shown in Formula (12):(12)c(n)=η

In Formula (12), η represents the weight coefficient of the effectiveness of the escape route for this node. Its value determines whether the escape route is effective. The larger the value, the more effective the escape route.

2.Improvement Strategy of Ant Colony Algorithm for Residual Information(1)Improved residual information

The concentration of residual information is the decisive factor for selecting the next point in the ant colony algorithm. If during the time interval from t to t + 1, m ants all start from the starting point, and after n time intervals, all the ants have traversed n paths, completing one cycle, then the residual information on each path is modified as follows in Formula (13):(13)$$\Delta \tau_{ij}^{k}=\left\{\begin{array}{l} \frac{2Q}{L_{k}},{\;L}_{k}<\min\{L_{1},\ldots ,L_{n}\}\\ \frac{Q}{L_{k}},{\;L}_{k}>=\min\{L_{1},\ldots ,L_{n}\}\\ 0,else \end{array}\right.$$

The above adaptive method for updating residual information based on path selection probability and adjusting the pheromone concentration of each path can effectively prevent redundancy and avoid getting trapped in local optima.

(2)Improved heuristic factor

In traditional ACO, the heuristic function only considers the Euclidean distance from the current node to the next node, lacking directional information towards the target point. This leads to poor pathing performance. In this paper, the Euclidean distance between the next node and the target point is added into the heuristic function, and the improved heuristic function is shown as Formula (14).
(14)$$\eta_{ij}=\frac{1}{d_{ij}+d_{jE}}$$

In Formula (14), $d_{ij}$ represents the equivalent distance of the path (i,j), and $d_{iE}$ is the Euclidean distance from the next node j to the target point E.

(3)Objective function

The objective function for fire escape route planning is shown as Formula (15).
(15)$$\begin{array}{l} r=\left\{i_{S},i_{1},i_{2},\ldots ,i_{E}\right\}\\ L_{SE}=\sum_{i=1}^{N}d_{r_{i},r_{i+1}}(t)\\ L_{opt}=\min(L_{SE}) \end{array}$$

In Formula (15), $i_{S}$ and $i_{E}$ represent the start and end points of the path, respectively. r denotes the sequence of path nodes, and N represents the evacuation path consisting of N sub-path segments. $L_{SE}$ is the equivalent distance of the feasible path from the starting point to the endpoint, and $L_{opt}$ is the equivalent length of the optimal path.

#### 3.1.4. Process of the Escape Route Planning Algorithm Based on IACO-A*

The algorithm process is as follows:Step 1: Initializing parameters. Establish the aggregate count of ants M and the upper limit of iterations N. Allocate $i_{S}$ starting values to the coefficients of pheromone concentration, heuristic function, residual pheromone coefficient, and the constant Q;Step 2: Choosing a route. Determine the likelihood of the k-th ant’s transition using specific formulas based on improved ACO. Employ the roulette tactic to select the subsequent node, guaranteeing a higher likelihood of choosing nodes with greater selection chances.Step 3: Constructing the path. Ascertain whether the present ant has arrived at its destination. Upon arriving at the endpoint, log the ant’s travel distance and modify the shortest existing route; if the ant hasn’t arrived at the endpoint, proceed to Step 2, choosing the subsequent node as previously mentioned, until the k-th ant arrives at the endpoint.Step 4: Ascertain if every one of the M ants has finished building their paths. Upon completion, modify the improved path pheromones; if not, revert to Step 2 and continue constructing the subsequent ant’s path.Step 5: Ascertain if the upper limit of iterations has been attained. Otherwise, go back to Step 2 and proceed with a fresh cycle of path choosing and building. Should the upper limit of iterations be surpassed, halt the computation and present the shortest route along with its length as the best solution for the objective function. Plans for the escape route have been finalized. See Figure 2.

### 3.2. The Fire Detection and Dynamic Path Planning Algorithm

By collecting environmental data and dividing it into the above fire stages, fire alarms and escape route planning can be conducted. The overall flowchart is shown in Figure 3.
Step 1: System initialization.Step 2: Generate the initial grid map.Step 3: The lower computer collects various data and uploads it to the LabVIEW upper computer.Step 4: Determine the fire situation based on various data. If the temperature is below 50 °C and the CO concentration is below 200, it is determined that no fire has occurred; if the temperature is between 50 °C and 80 °C and the CO concentration is between 200 and 2000, it is determined to be the early stage of a fire, and an early-stage fire map is generated; if the temperature is above 80 °C or the CO concentration is above 2000, it is determined to be the mid-to-late stage of a fire, and a mid-to-late-stage fire map is generated.Step 5: Implement the evacuation route planning algorithm to map out escape routes for different stages of the fire based on the map generated in Step 4.Step 6: Finish.

### 3.3. Design of the Upper Computer

The front panel of the LabVIEW host computer includes sliding function pages, a data acquisition area, and a path planning area, as shown in Figure 4.

#### 3.3.1. Sliding Pages

Figure 4 shows the set up of three horizontally arranged sliding buttons. By clicking the buttons, you can slide between pages. The user employs the tab control to create the pages corresponding to each button. The three pages correspond to the stages of no fire, early fire, and mid-to-late fire.

#### 3.3.2. Data Collection Area

The data collection area includes a data setting module, a data display module, and a data alarm module. The data setting module allows for serial port settings, temperature settings, and CO gas concentration settings. The data display module includes real-time data display and waveform graph display. The dashboard shows the real-time CO gas concentration, while the thermometer displays the real-time temperature. The data alarm module consists of two Boolean lights that indicate temperature and CO gas concentration. By processing and analyzing the data, it determines whether the data is within the normal range. If within the normal range, the Boolean lights remain off; if not, the Boolean lights turn on, activating the escape route planning function. Depending on the stage of fire development, it plans escape routes for the early and later stages of the fire.

#### 3.3.3. Path Planning Area

The escape route planning function needs to be implemented in coordination with MATLAB. In the program panel, a MATLAB program window is established and the escape route code is written based on the ant colony algorithm. In this code, ‘data_map’ is the map creation function, ‘ACO’ is the route planning function, and ‘Rx’ and ‘Ry’ are the horizontal and vertical coordinate parameters of each point on the route. The program diagram for the rear panel is shown in Figure 5.

## 4. Experimental Design and Verification

### 4.1. Setting Up the Test Environment

To demonstrate the reasonableness and reliability of the method proposed in this paper, a simulation experiment was conducted based on an indoor fire scenario in a school teaching building. The walls, columns, and stairs of the experimental teaching building were set to be inert and did not participate in the fire combustion. Initial environmental parameters were also set as follows: the ambient temperature was 26 °C, the pressure was standard atmospheric pressure, the relative humidity was 40%, and the ventilation wind speed was 2 m/s. Additionally, nine sensor nodes were reasonably arranged within the building, located at four safety exits and at intersections of the passages, to collect temperature and carbon monoxide concentration data. The simulation was carried out in the MATLAB environment, using a grid size of 20×20 to simulate the real scene, with the unit being meters. The initial map is shown in Figure 6. In the figure, black squares represent obstacles, white squares represent passable points, and the locations of sensor nodes 1~9 and escape EXIT-1~4 are marked in yellow and red, respectively. A desktop computer equipped with an Intel Core i7-7700 3.6 GHz CPU was used to conduct simulation experiments on the Windows 10 operating system with MATLAB 2023 software.

### 4.2. Simulation Testing of Escape Route Planning

According to Table 1, the fire situation is divided into three stages: no fire, early-stage fire, and mid-to-late-stage fire. Sensor data are collected at 0.25 h intervals. Based on the sensor data, the fire situation and spread trend are determined.

The ant colony algorithm is then used to iterate and optimize the shortest accessible path. The parameters for the algorithm are set as shown in Table 2. The path planning results are as follows:

#### 4.2.1. Path Planning for the Early Stage of Fires

Analysis of fire incidents

The data collected from each monitoring point in the test environment between 8:00 a.m. and 10:00 a.m. are shown in Table 3. Starting from 9:00 a.m., significant fluctuations appeared in the data from monitoring points 1 and 5, indicating that a fire occurred at these two points. At this time, the temperature ranged from 55 °C to 80 °C, and the CO concentration ranged from $300\times 10^{-6}$ to $2000\times 10^{-6}$, indicating that the fire was in its early development stage. The areas with fire incidents are highlighted in yellow in Table 3. At this point, the fire was in a small-scale burning state, with a slow CO diffusion rate and low concentration, covering a small area. The temperature and CO concentration were both below the maximum tolerance levels for the human body and did not yet pose a threat to human health. The approximate locations of the fire source were at (5, 10) and (7, 5).

According to the fire hazard classification standards in Table 1, a comprehensive analysis of the sensor data and the fire spread trend was conducted. The fire spread trend graph for the early stages of the fire development is shown in Figure 7.

2.Escape route planning

According to the trend of the fire spreading shown in Figure 5, there are severe fire conditions at EXIT-1 and EXIT-4. Therefore, the designated escape routes are EXIT-2 and EXIT-3. The updated grid map is shown in Figure 8, where the black grid sections represent impassable areas, and the white grid sections represent safe passage areas.

Based on the updated grid map, the escape routes were planned, and the results are shown in the figure. Figure 9a–c depicts the escape route planning results for trapped individuals at three different locations, with exits EXIT-2 and EXIT-3 as the destinations. The escape routes to EXIT-2 are marked in blue, and the escape routes to EXIT-3 are marked in red. In Figure 9a, the trapped individual starts from the upper left corner of the map, with coordinates (3, 17). The length of path 1 is 20.071, and the length of path 2 is 8.414. In Figure 9b, the trapped individual starts from the lower right corner of the map, with coordinates (13, 4). The length of path 1 is 9.243, and the length of path 2 is 17.828. In Figure 9c, the trapped individual starts from a position slightly to the right of the center of the map, with coordinates (12, 10). The length of path 1 is 8.414, and the length of path 2 is 9.828.

#### 4.2.2. Initial Path Planning for Fires

Analysis of fire incidents

The data collected from each monitoring point in the environment being tested between 10:00 a.m. and 12:00 a.m. are shown in Table 4. The temperature and CO concentration at monitoring points 1 and 5 continued to rise, reaching temperatures above 80 °C and CO concentrations above $2000\times 10^{-6}$, indicating that the fire had entered the middle to late stages of development. The areas with fire incidents are highlighted in yellow in Table 4. At this time, the fire gradually formed a massive convection with outdoor air through doors and windows, further expanding the area covered by the fire in the corridor. The fire temperature and CO concentration continued to rise, exceeding the maximum tolerable levels for the human body, posing a severe threat to people’s health.

Compared to earlier data, there is a significant change at monitoring point 9, indicating that the fire is gradually spreading to the location of monitoring point 9.

The trend map of fire spread in the middle and later stages of the fire development is drawn as shown in Figure 10.

2.Escape route planning

According to the fire spread situation shown in Figure 11, the updated grid map is shown in Figure 9. It can be seen that, compared to the grid map in the early stages of the fire, the mid-to-late-stage grid map shows a larger black area in the lower left corner.

The escape routes are planned based on the updated grid map, with the results shown in Figure 12a–c. The coordinates of the trapped individuals and the escape exits are the same as those in the early stage of fire experiments. In Figure 12a, the length of path 1 is 20.071, and the length of path 2 is 8.414. In Figure 12b, the length of path 1 is 9.243, and the length of path 2 is 20.071. In Figure 12c, the length of path 1 is 8.414, and the length of path 2 is 16.071.

From the simulation results in Figure 9 and Figure 12, it is evident that the proposed IACO-A* algorithm possesses excellent path optimization capabilities, finding the optimal escape route across various terrain environments.

### 4.3. Testing of Upper Computer

After the path planning algorithm runs successfully, it needs to be debugged with the LabVIEW upper computer, configuring the parameters required for the serial port module on the rear panel where the VISA resource name is set to COM1 port, baud rate to 9600 bps, data bits to 8, and parity bit to “no parity”. The serial port assistant simulator is used to transmit simulated data, testing the data monitoring and path planning functions of the upper computer. Three scenarios are selected for data testing and path planning.

#### 4.3.1. Upper Computer’s Test for Early Stage of the Fire

As shown in Figure 11, the upper computer displays the real-time temperature as 70 °C and the real-time CO concentration as 750 × 10^−6^, exceeding the upper limits for temperature and CO concentration. The temperature alarm light and CO concentration alarm light are on. The “Path Planning” button is clicked, and after a few seconds the path planning results will appear. As shown in Figure 13a,b, in the early stage of the fire the path planning results of the upper computer start from the coordinate (3, 17) and end at EXIT-2 and EXIT-3, respectively.

#### 4.3.2. Upper Computer’s Test for Mid-to-Late Stage of the Fire

As shown in Figure 14, the host computer displays the real-time temperature as 85 °C and the real-time CO concentration as 2020 × 10^−6^, exceeding the upper limits for temperature and CO concentration. The temperature alarm light and the CO concentration alarm light are on. The ‘Path Planning’ button is clicked, and after waiting a few seconds the path planning results will appear. As shown in Figure 14a,b, in the mid-to-late stage of the fire, the path planning results of the upper computer start from the coordinate (13, 4) and end at EXIT-2 and EXIT-3, respectively.

#### 4.3.3. Upper Computer’s Test in Case of High CO Levels

As shown in Figure 15, the upper computer displays the real-time temperature as 50 °C and the real-time CO concentration as 700 × 10^−6^. The temperature is within the normal range, but the CO concentration exceeds the normal range, triggering the CO concentration alarm light. The ‘Path Planning’ button is clicked, and after a few seconds the path planning results will appear. As shown in Figure 15a,b, when the CO concentration is too high, the path planning results of the host computer start from coordinate (12, 10) and end at EXIT-2 and EXIT-3, respectively.

### 4.4. Escape Route Planning in Case of Node Data Abnormality

In the complex environment of a fire scene, unexpected anomalies in sensor nodes are unavoidable, and the loss of real-time data from these nodes can severely impact the planning of subsequent escape routes. Therefore, it is necessary to address the potential risks of data loss caused by malfunctioning nodes by detecting and excluding the data from these abnormal nodes.

Taking the later stage of fire growth as an example, with the rapid spread of the fire the corridors are in a high-temperature complex environment, as shown in Figure 16. At this time, the temperature and CO concentration at monitoring point 9 significantly decrease, indicating that data collection at this node is experiencing issues.

Based on the above situation, an abnormal mechanism handling plan is initiated to exclude the data of abnormal nodes, and the LSTM neural network is used for data prediction. The fire trend chart after the abnormal node data repair is shown in Figure 17, and the escape route is planned accordingly. The results are shown in Figure 12a–c.

As can be seen from Figure 17, the temperature and CO concentration in the previously marked abnormal data areas have been corrected, and their actual conditions have returned to what was described in Section 4.2.2.

### 4.5. Performance Comparison

Figure 18 shows the comparison of convergence between the original ACO algorithm and the proposed IACO-A* algorithm under the fire situation model in the later stages of a fire, with the point (3, 17) as the starting point and EXIT-3 as the escape exit. It can be observed that in multiple experiments, the original ACO algorithm converged at the 100th iteration, with the shortest path being 9.243 m. On the other hand, the IACO-A* algorithm converged at the 73rd and 80th iterations in three experiments, with the shortest path being 8.414 m. This shows that the improved algorithm yields shorter path lengths and better convergence effects.

Furthermore, a comparison of the path planning of the two algorithms for different trapped personnel positions in the early and mid-later stages of a fire is shown in Table 5.

As can be seen from Table 5, the improved ant colony algorithm significantly enhances the convergence speed and reduces the number of iterations, effectively avoiding the problem of optimization algorithms easily getting trapped in local optima. Compared to the original ant colony algorithm, the planned escape routes are shortened by an average of 17.1%. It is evident that the improved algorithm proposed in this paper can achieve better path planning results in a shorter time. The optimized improved algorithm is more suitable for the evacuation of people in complex building fires.

## 5. Conclusions

This paper proposes a fire alarm monitoring and escape route planning system based on an improved ant colony algorithm, conducting indoor fire safety escape simulation experiments with the teaching building of a certain school as the background. The following conclusions were drawn from the analysis and testing:(1)Compared to the ACO model, the IACO-A* model considers the real-time dispersion of fire products, allowing the evacuation paths to reflect the impact of fire spread on path planning. Thus, evacuation routes can be adjusted in real-time according to the development of the fire, enabling effective avoidance of hazardous areas under different fire scenarios. This further ensures the safety of evacuees compared to the ACO algorithm and can be applied in practical fire escape path planning.(2)The improved algorithm based on ‘A* + Ant Colony’ yields better results, converges faster, and demonstrates good real-time performance.(3)The abnormal node data correction method addresses issues of false alarms, missed reports, and data loss caused by abnormal sensor node operation in high-temperature fire environments. By integrating a phased path planning method based on fire spread trends, it can more accurately simulate actual fire scenarios. This helps to mitigate the impact of issues such as incomplete or inaccurate real-time monitoring data from nodes, improving the timeliness and reliability of escape route planning.(4)The fire monitoring and escape route monitoring system built on the LabVIEW platform is versatile. This effectively combines the big data of fire detection terminals with artificial intelligence technology, helping to enhance the intelligence level of building fire safety management. This can serve as a reference for indoor fire alarm monitoring technology.

In future research, we will consider fire monitoring and path planning in complex environments with multiple exits and multi-story buildings. We will introduce a fire simulation system to more accurately simulate on-site fire conditions. Additionally, the path planning in this paper is based on an ideal constant speed scenario. In future research we will consider how environmental factors affect human movement speed and survival conditions, and select appropriate indicators to further explore this field.

## Figures and Tables

**Figure 1 sensors-24-06438-f001:**
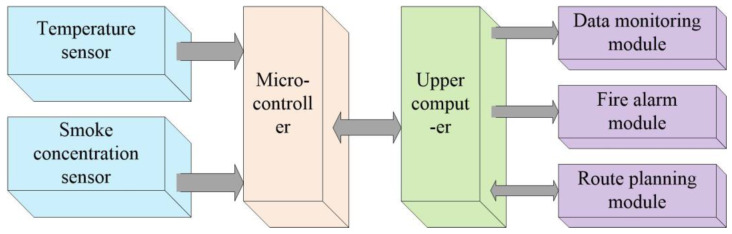
System architecture diagram.

**Figure 2 sensors-24-06438-f002:**
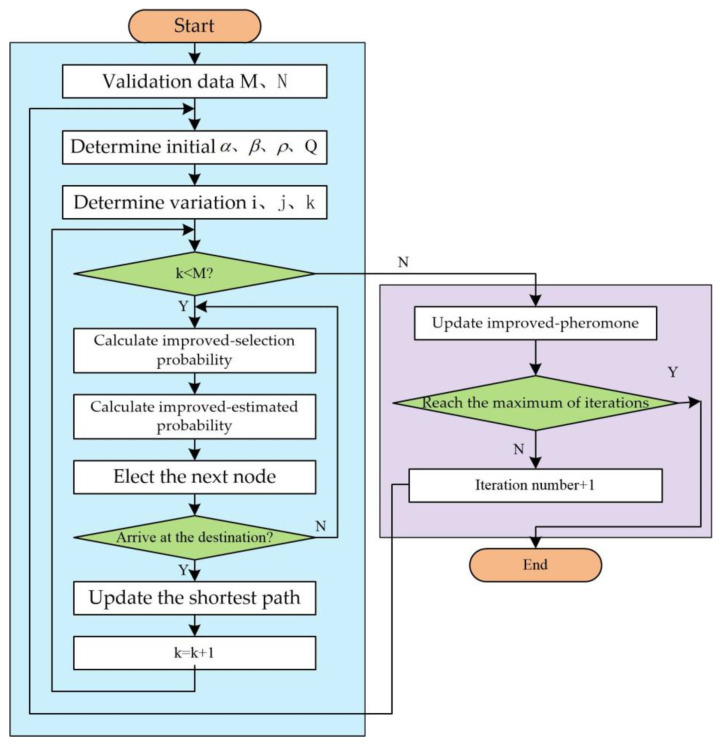
IACO-A*algorithm.

**Figure 3 sensors-24-06438-f003:**
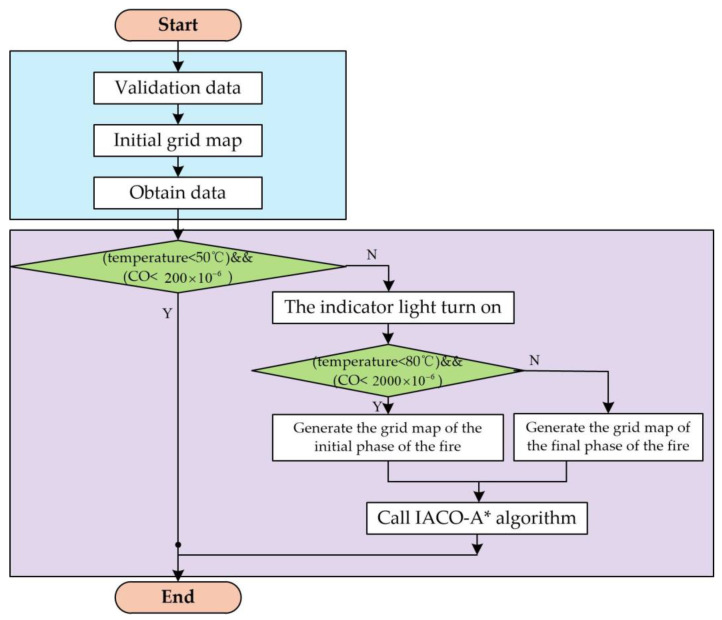
The fire detection and dynamic path planning algorithm.

**Figure 4 sensors-24-06438-f004:**
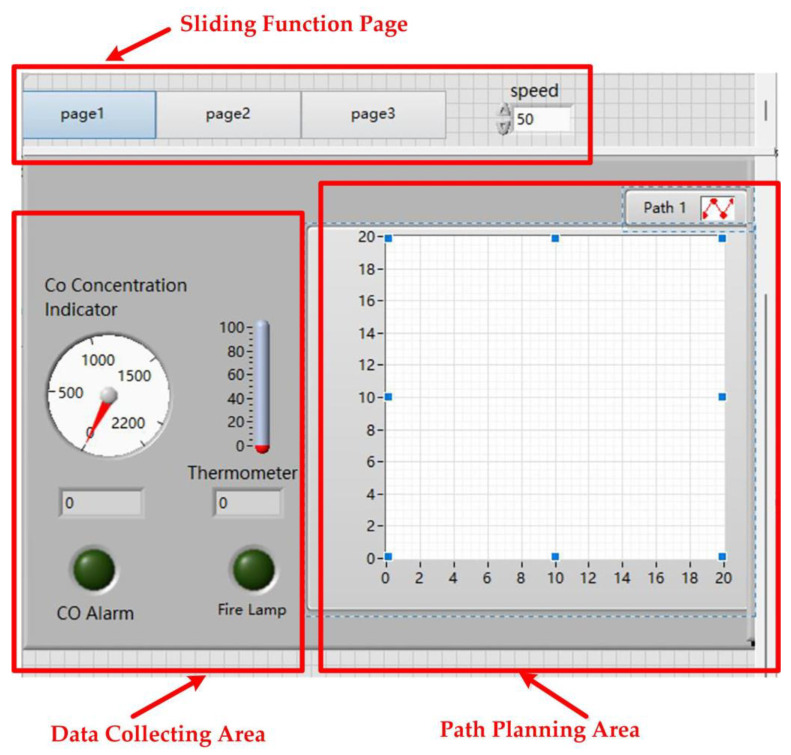
The front panel of the LabVIEW host computer.

**Figure 5 sensors-24-06438-f005:**
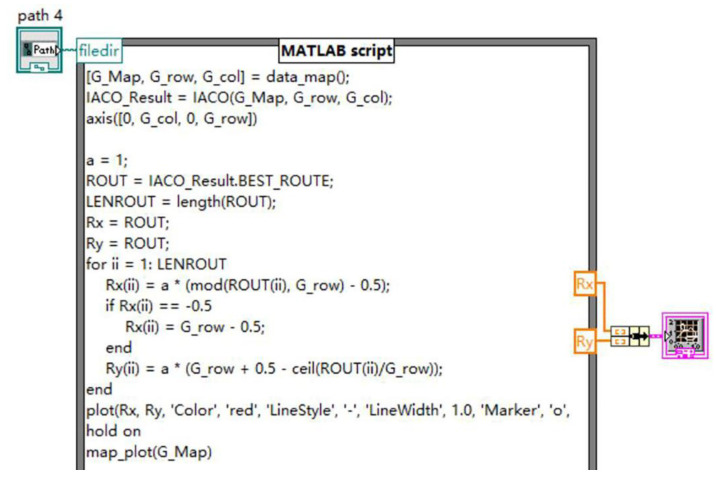
Path planning program panel.

**Figure 6 sensors-24-06438-f006:**
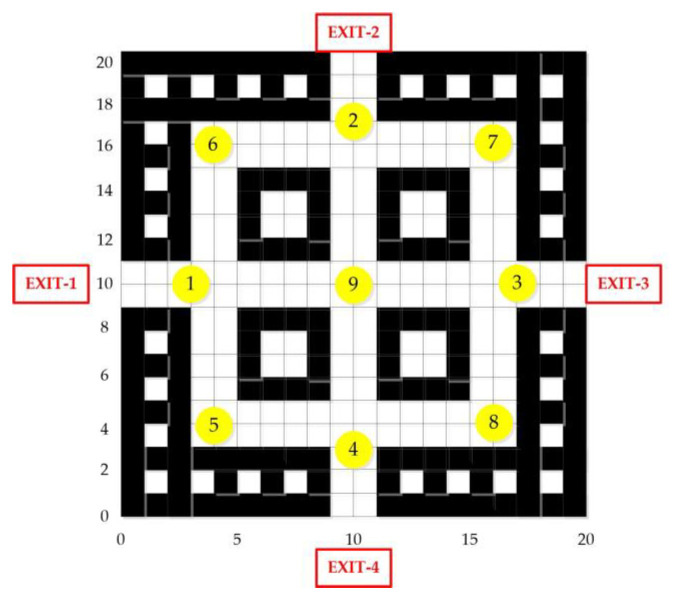
Initial grid map.

**Figure 7 sensors-24-06438-f007:**
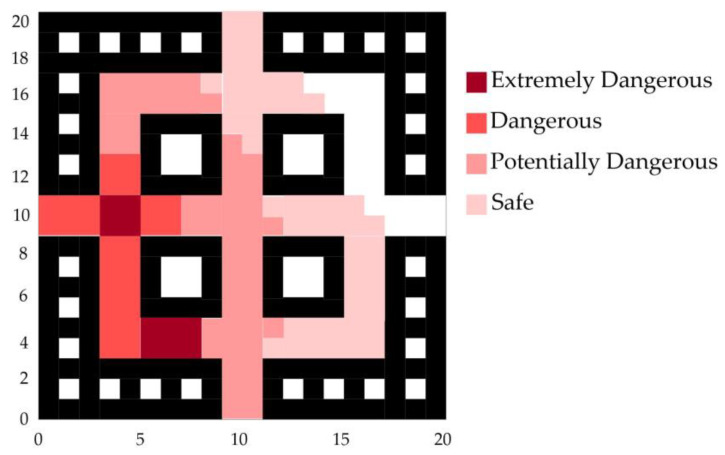
Fire spread trend graph for the early stages of fire.

**Figure 8 sensors-24-06438-f008:**
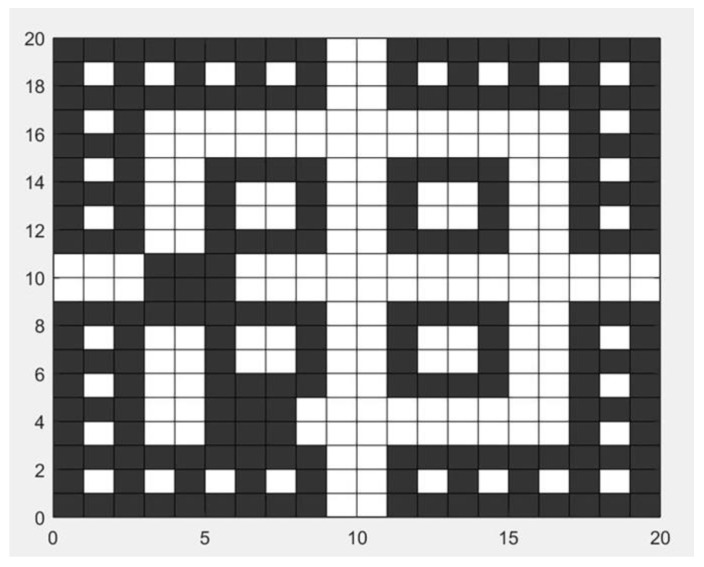
Grid map of the early stage of fire.

**Figure 9 sensors-24-06438-f009:**
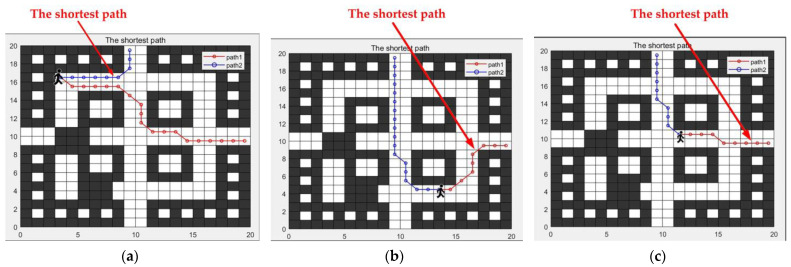
Escape routes in the early stages of fire. (**a**) The escape routes start from coordinates (3, 17). (**b**) The escape routes start from coordinates (13, 4). (**c**) The escape routes start from coordinates (12, 10).

**Figure 10 sensors-24-06438-f010:**
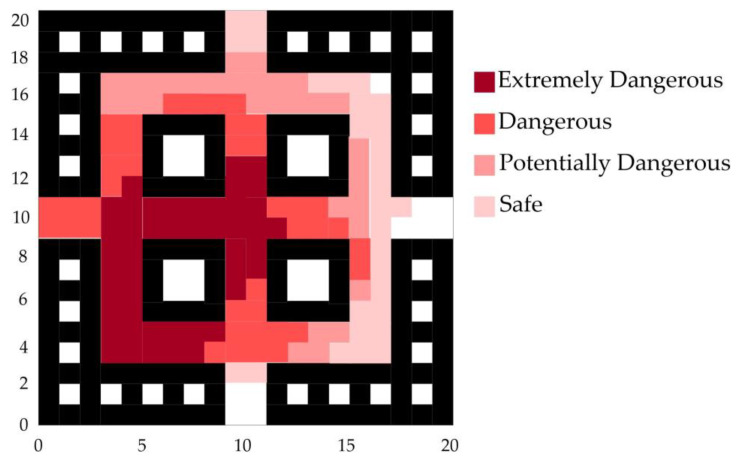
Fire spread trend graph for the middle and later stages of fire.

**Figure 11 sensors-24-06438-f011:**
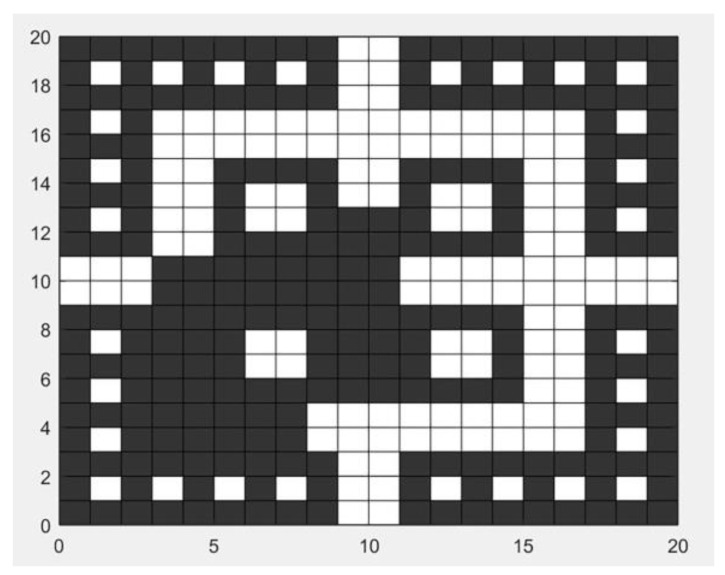
Grid map of the middle and later stages of fire.

**Figure 12 sensors-24-06438-f012:**
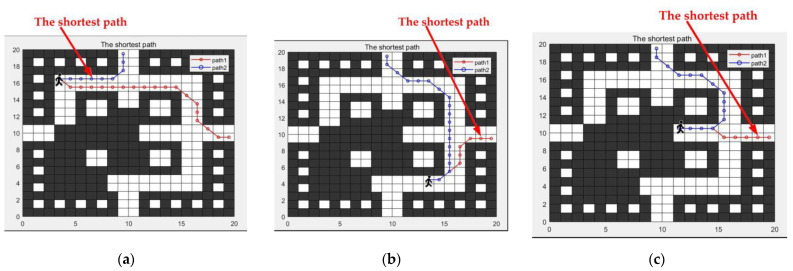
Escape routes in the middle and later stages of fire. (**a**) The escape routes start from coordinates (3, 17). (**b**) The escape routes start from coordinates (13, 4). (**c**) The escape routes start from coordinates (12, 10).

**Figure 13 sensors-24-06438-f013:**
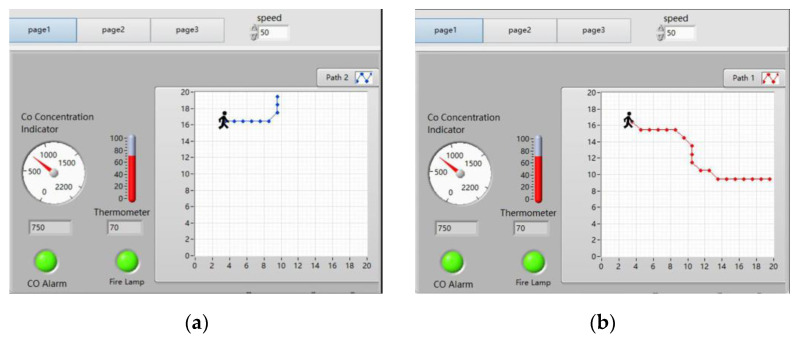
Upper computer’s path planning results in the early stage of the fire. (**a**) The result of the upper computer’s path planning from the starting point at coordinates (3, 17) to EXIT-2. (**b**) The result of the upper computer’s path planning from the starting point at coordinates (3, 17) to EXIT-3.

**Figure 14 sensors-24-06438-f014:**
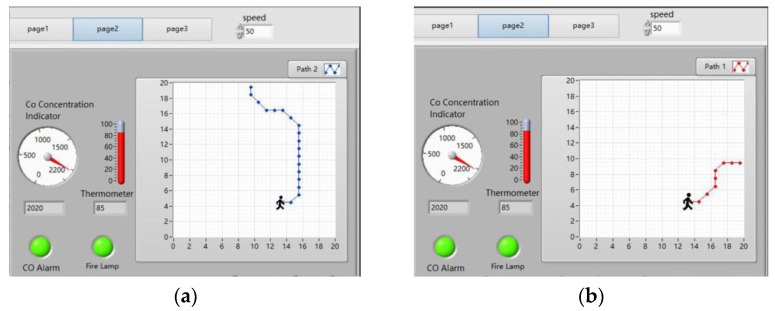
Upper computer’s path planning results in the mid-to-late stage of the fire. (**a**) The result of the upper computer’s path planning from the starting point at coordinates (13, 4) to EXIT-2. (**b**) The result of the upper computer’s path planning from the starting point at coordinates (13, 4) to EXIT-3.

**Figure 15 sensors-24-06438-f015:**
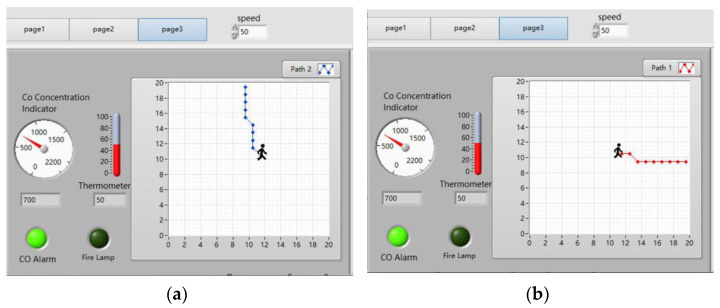
Upper computer’s path planning results in case of high CO levels. (**a**) The result of the upper computer’s path planning from the starting point at coordinates (12, 10) to EXIT-2. (**b**) The result of the upper computer’s path planning from the starting point at coordinates (12, 10) to EXIT-3.

**Figure 16 sensors-24-06438-f016:**
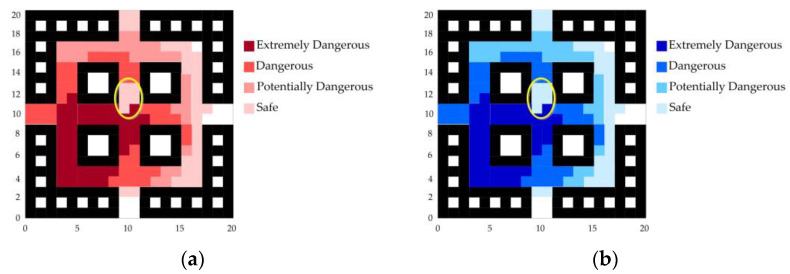
Fire spread trend graph in case of node data abnormality. (**a**) Temperature diffusion graph in case of node data abnormality. (**b**) CO concentration diffusion graph in case of node data abnormality.

**Figure 17 sensors-24-06438-f017:**
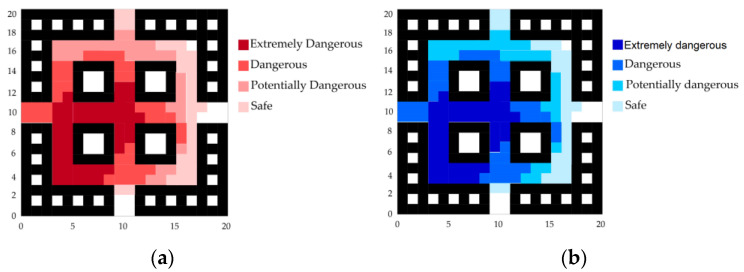
Fire spread trend graph in case of node data abnormality. (**a**) Temperature spread trend graph after abnormal node data repair. (**b**) CO concentration spread trend graph after abnormal node data repair.

**Figure 18 sensors-24-06438-f018:**
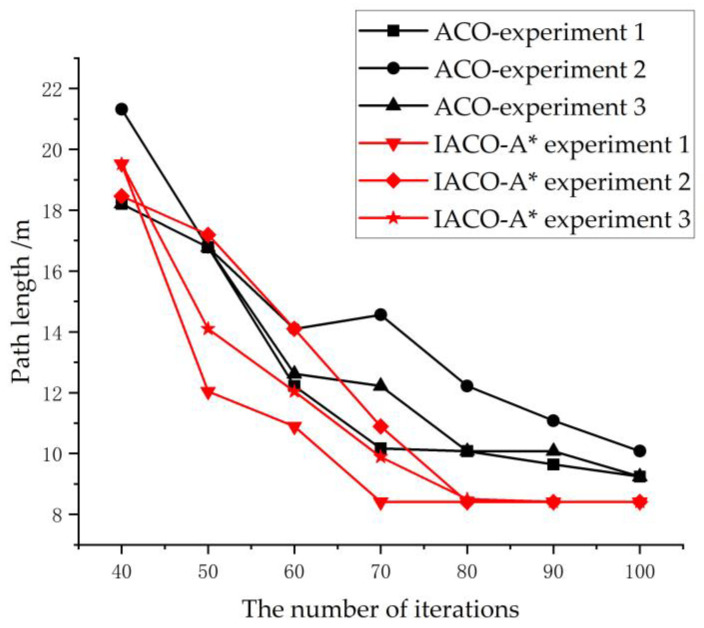
Comparison of convergence effects with different iteration counts.

**Table 1 sensors-24-06438-t001:** Classification of fire hazard levels and stages.

Level of Fire	Stage of Fire	Temperature/°C	Concentration/ppm
Safe	No fire	<42	<55
Potential danger	No fire	42~50	50~200
Dangerous/numbing to humans	Early stage	50~80	200~2000
Extremely dangerous/fatal	Later stage	>80	>2000

**Table 2 sensors-24-06438-t002:** Route planning parameter settings.

M	α	β	ρ	Q	Iter
50	1	7	0.3	1	100

**Table 3 sensors-24-06438-t003:** Sensor data in the early stage of fire.

Sensor Serial Number	Sensor Data	Time/a.m.								
		8:00	8:15	8:30	8:45	9:00	9:15	9:30	9:45	10:00
No. 1	Temperature	26	26	28	29	54	55	58	60	63
	CO concentration	30	30	31	35	300	310	500	595	753
No. 2	Temperature	26	26	27	30	39	39	40	40	41
	CO concentration	30	30	30	31	42	43	46	45	47
No. 3	Temperature	26	26	26	27	30	30	33	33	32
	CO concentration	32	32	34	35	38	38	37	38	39
No. 4	Temperature	26	26	25	26	48	48	46	47	47
	CO concentration	31	30	30	36	112	125	150	156	190
No. 5	Temperature	26	26	27	31	60	61	65	68	70
	CO concentration	30	30	30	30	315	408	601	780	830
No. 6	Temperature	26	26	26	28	43	44	46	46	48
	CO concentration	34	34	31	44	82	88	126	162	173
No. 7	Temperature	26	26	24	26	27	27	38	38	37
	CO concentration	30	30	32	32	34	34	351	35	26
No. 8	Temperature	27	27	28	29	32	32	34	34	34
	CO concentration	34	34	31	34	42	43	45	45	46
No. 9	Temperature	28	28	29	29	42	42	44	45	47
	CO concentration	35	35	33	37	61	70	77	90	92

**Table 4 sensors-24-06438-t004:** Sensor data in the mid-to-late stages of fire.

Sensor Serial Number	Sensor Data	Time/a.m.								
		10:00	10:15	10:30	10:45	11:00	11:15	11:30	11:45	12:00
No. 1	Temperature	63	64	68	68	71	76	74	76	78
	CO concentration	753	802	810	865	885	901	920	948	960
No. 2	Temperature	41	43	42	42	45	47	47	47	48
	CO concentration	47	54	120	140	143	156	162	169	184
No. 3	Temperature	32	32	33	34	32	32	31	33	29
	CO concentration	39	39	38	38	35	34	32	32	30
No. 4	Temperature	47	48	52	57	57	63	62	62	60
	CO concentration	190	214	230	296	300	320	327	379	389
No. 5	Temperature	70	75	76	84	83	92	89	89	94
	CO concentration	830	997	2001	2015	2046	2040	2031	2024	2021
No. 6	Temperature	48	49	52	59	67	69	68	67	64
	CO concentration	173	179	230	332	350	387	452	448	441
No. 7	Temperature	24	28	36	36	38	40	38	38	37
	CO concentration	36	39	43	44	48	50	51	52	50
No. 8	Temperature	34	41	45	47	48	50	47	45	45
	CO concentration	46	83	93	118	123	135	156	149	143
No. 9	Temperature	47	58	83	89	85	86	90	92	93
	CO concentration	92	2004	2060	2059	2049	2051	2062	2059	2052

**Table 5 sensors-24-06438-t005:** Comparison of results between the improved ACO and the original ACO.

Starting Point			(3, 17)		(13, 4)		(12, 10)	
Emergency Exit Number			EXIT-2	EXIT-3	EXIT-2	EXIT-3	EXIT-2	EXIT-3
Early stage of fire	Original ACO	Path length/m	24.556	9.243	9.512	20.071	9.243	11.205
		Number of iterations	100	100	100	100	98	100
		Runtime/s	16.64	16.32	16.71	16.43	16.56	16.34
	Improved ACO	Path length/m	20.071	8.414	9.243	17.828	8.414	9.828
		Number of iterations	70	71	66	80	70	84
		Runtime/s	12.32	12.78	12.09	12.54	12.70	12.33
Mid-to-late stage of fire	Original ACO	Path length/m	24.314	9.243	9.512	25.728	9.243	20.899
		Number of iterations	100	100	100	100	100	100
		Runtime/s	16.04	16.67	16.64	16.32	16.20	15.44
	Improved ACO	Path length/m	20.071	8.414	9.243	20.071	8.414	16.071
		Number of iterations	71	73	70	78	76	80
		Runtime/s	12.51	12.54	12.65	12.41	12.18	12.21

## Data Availability

Data are contained within the article.

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
