# Peer review of "Design of Intelligent Firefighting and Smart Escape Route Planning System Based on Improved Ant Colony Algorithm"

_sensors, 2024, doi:10.3390/s24196438_

Round 1
Reviewer 1 Report
Comments and Suggestions for Authors
This paper uses the improved ant colony optimization to design the firefighter and smart escape route. The topic of this paper is interesting for engineering applications. My comments are below:
1) This paper presents only one plan of the building. Is it possible to simulate more complicated planning?
2) Generally, the building always has many levels, is this proposed to be applied with more level buildings? This suggestion will be important for practical use because this is a more complicated problem and the authors must implement this condition's cost and basic infrastructure.
3) Why the improved ant colony optimization is selected as the optimizer for this work?
4) The optimization problem and constraint details must be explained in mathematical form.
5) The convergence of the improved ant colony optimization must be shown and what happens when the optimum solution gives a different solution? Because the solution of ACO will be changed for every run.
Comments on the Quality of English LanguageMinor editing of English language required.
Author Response
Dear Editors and Reviewers:
Thank you for your letter and for the reviewers’ comments concerning our manuscript entitled “Design of an Intelligent Firefighting and Smart Escape Route Planning System Based on Improved Ant Colony Algorithm” (Manuscript Number: sensors-3201088). Those comments are all valuable and very helpful for revising and improving our paper, as well as the important guiding significance to our researches. We have studied comments carefully and have made correction which we hope meet with approval.
Revised portion are marked in yellow in the paper. The main corrections in the paper and the responds to the reviewers’ comments are as flowing:
Response to Reviewer #1:
[Comment 1]This paper presents only one plan of the building. Is it possible to simulate more complicated planning?
Reply:We are very grateful to the reviewer for pointing out the shortcomings in the complexity of our simulation model, and we have added additional explanations in the conclusion section.
How the paper is modified: The following is the description we added in the conclusion section:
|
Before modification
|
|
After modification
|
[Comment 2]Generally, the building always has many levels, is this proposed to be applied with more level buildings? This suggestion will be important for practical use because this is a more complicated problem and the authors must implement this condition's cost and basic infrastructure.
Reply: We are very grateful to the reviewers for pointing out the shortcomings in our manuscript. In the outlook for future work in the conclusion section, the prospect of applying this method to more multi-story buildings was discussed. The research scenario of this paper is focused on large multi-story indoor buildings, which, based on the fact that most large indoor buildings are already equipped with fire alarm systems, provides the necessary infrastructure for this application prospect.
How the paper is modified: A description of the improvements in our abstract section follows:
|
Before modification
|
|
After modification
|
[Comment 3] Why the improved ant colony optimization is selected as the optimizer for this work?
Reply: We are very grateful to the reviewers for pointing out our insufficient explanation regarding the reasons behind the choice of algorithms.We have supplemented the discussion with the advantages and limitations of the ant colony algorithm, as well as the motivation behind the improvements proposed in this paper.
How the paper is modified: At the beginning of Section 3.1, reasons for choosing the improved ACO algorithm as the path planning optimizer for this paper are added:
|
After modification |
[Comment 4] The optimization problem and constraint details must be explained in mathematical form.
Reply: We are very grateful to the reviewers for pointing out the shortcomings in our mathematical description of the algorithm model. We have added formulas for the improved algorithm's constraints, objective functions, and corresponding explanatory text.
How the paper is modified:
1)In Section 3.1.2, formulas 5-7 and additional theoretical explanations were added to explain the fire spread model;
|
Before modification |
|
After modification |
|
After modification |
2)The previous section 3.1.2 has now been moved to section 3.1.3.
|
Before modification |
|
After modification |
3)Move the 'Analysis of Fire Factors' section from 3.2.1 to 3.1.2
|
Before modification |
|
After modification |
- In Section 3.1.3-1, formulas 9-12 and additional theoretical explanations were added to further detail the improvements in the evaluation and loss functions in the path planning algorithm after introducing the A* algorithm;
|
After modification |
5)in Section 3.1.3-2, formula 14 was added to explain the heuristic function of the improved algorithm, and formula 15 was added to explain the objective function of the algorithm in this paper.:
|
Before modification |
|
After modification |
|
After modification |
There are 15 formulas in the revised version of this document.
[Comment 5]The convergence of the improved ant colony optimization must be shown and what happens when the optimum solution gives a different solution? Because the solution of ACO will be changed for every run.
Reply: We are very grateful to the reviewer for pointing out our insufficient discussion regarding the algorithm's convergence, as well as our lack of consideration for the differing results in multiple experiments.
How the paper is modified:
1)In Section 4.5, Figure 16 and the related analytical text were added to demonstrate the convergence of the improved algorithm across multiple experiments.
|
After modification
|
The experiments show that the algorithm consistently presents the optimal solution in each instance.
According to the reviewers’ comments, we have made extensive modifications to our manuscript and supplemented extra data to make our results convincing. Thank you again for your positive comments and valuable suggestions to improve the quality of our manuscript.
We appreciate for editors and reviewers’ warm work earnestly, and hope that the correction will meet with approval.
Yours sincerely,
Song Yafei
September 18, 2024

Reviewer 2 Report
Comments and Suggestions for Authors
General comments
We thank the authors for their commendable efforts. The manuscript provides a great insight into the intelligent fire monitoring and dynamic evacuation route planning for indoor environments, with the goal of addressing shortages in present evacuation systems using an upgraded algorithm.
Specific Comments
Title:
- The title effectively captures the essence of the research, which is discussing the development and implementation of a smart firefighting and escape route planning system based on an improved ant colony algorithm.
Abstract:
- The abstract is meticulously organized and offers a succinct overview of the study's aims, methodologies, and significant discoveries, but it might need an overview detailed of the ant colony technique modifications.
Introduction:
- The introduction adeptly establishes the context by providing a comprehensive description of the difficulties of fire monitoring and evacuation in vast interior settings, highlighting the importance of synchronized fire alarm systems.
- Some sections of the paper, especially the introductory section, might benefit from clearer and more concise language.
Methodology:
- The approach is comprehensive, providing full information about the system's design, the upgraded ant colony algorithm, and the integration of LabVIEW and MATLAB for real-time data visualization.
- Enhancing the study could yield more technical details about the parameter adjustments and the proposed method's computing cost.
Results and Discussion:
- The results are meticulously structured, with several parts that encompass the simulation environment and parameter configurations. Utilizing MATLAB for simulation and validation enhances the study's robustness.
- Figure 7: The readability of the legend could be improved by employing unique colours or patterns to distinguish between various methods.

Author Response
Dear Editors and Reviewers:
Thank you for your letter and for the reviewers’ comments concerning our manuscript entitled “Design of an Intelligent Firefighting and Smart Escape Route Planning System Based on Improved Ant Colony Algorithm” (Manuscript Number: sensors-3201088). Those comments are all valuable and very helpful for revising and improving our paper, as well as the important guiding significance to our researches. We have studied comments carefully and have made correction which we hope meet with approval.
Revised portion are marked in yellow in the paper. The main corrections in the paper and the responds to the reviewers’ comments are as flowing:
Response to Reviewer #2:
[Comment 1] Title:The title effectively captures the essence of the research, which is discussing the development and implementation of a smart firefighting and escape route planning system based on an improved ant colony algorithm.
Reply: We are very grateful to the reviewers for our paper.
[Comment 2] Abstract:The abstract is meticulously organized and offers a succinct overview of the study's aims, methodologies, and significant discoveries, but it might need an overview detailed of the ant colony technique modifications.
Reply: We are very grateful to the reviewers for pointing out the shortcomings in our manuscript.
How the paper is modified: In abstract,Added details on the improvements of the ant colony algorithm.
|
Before modification |
|
After modification |
[Comment 3] Introduction:The introduction adeptly establishes the context by providing a comprehensive description of the difficulties of fire monitoring and evacuation in vast interior settings, highlighting the importance of synchronized fire alarm systems.
Some sections of the paper, especially the introductory section, might benefit from clearer and more concise language.
Reply: We are very grateful to the reviewer for pointing out our shortcomings in terms of language expression.How the paper is modified: We add a comparison table of the prediction errors in the experimental section, and add the table and its description as shown below:
How the paper is modified: We have already re-examined the entire text and revised the wording to ensure it is as clear and accurate as possible.
[Comment 4] Methodology:The approach is comprehensive, providing full information about the system's design, the upgraded ant colony algorithm, and the integration of LabVIEW and MATLAB for real-time data visualization.
Enhancing the study could yield more technical details about the parameter adjustments and the proposed method's computing cost.
Reply: We are very grateful to the reviewer for pointing out the lack of the technical details.
How the paper is modified: In section 4.5,added figure 16 to show the convergence of different algorithms over multiple experiments and iterations.The time duration of each experiment has been added in Table 5 to examine the real-time performance of the algorithm.
|
Before modification |
|
After modification |
[Comment 5] Results and Discussion:The results are meticulously structured, with several parts that encompass the simulation environment and parameter configurations. Utilizing MATLAB for simulation and validation enhances the study's robustness.
Figure 7: The readability of the legend could be improved by employing unique colours or patterns to distinguish between various methods.
Reply: We are very grateful to the reviewer for pointing out the lack of the paper.
How the paper is modified: Arrows and text have been added in Figure 7 to emphasize the optimal result of each simulation.
|
Before modification |
|
After modification
|
According to the reviewers’ comments, we have made extensive modifications to our manuscript and supplemented extra data to make our results convincing. Thank you again for your positive comments and valuable suggestions to improve the quality of our manuscript.
We appreciate for editors and reviewers’ warm work earnestly, and hope that the correction will meet with approval.
Yours sincerely,
Song Yafei
September 18, 2024

Reviewer 3 Report
Comments and Suggestions for Authors
The manuscript should be revised seriously based on the comments below:
1) Only about 20 references were used for literature review. Please expand your overview by discussing at least 30 references.
2) The two first paragraphs in Introduction do not indicate the motivation of the study. Please point out studied objects, objective functions, and constraints. In addition, Please say why you choose the topic.
3) Algorithm 1 and Algorithm 2 must be named. And please use flowchart for them to clarify their applications.
4) The study lacks equations, only 6 equations for a study to be published in SCIE journal. This not satisfy the standard for a high-quality work as your paper.
5) In figure 3, you show many equations but you did not clarify them in body text. Please put them in body text and clarify them by explain their purpose, define each symbol and cite reference for each.
6) Check your typos and English: "Figure1, Figure2, Figure3", check the caption of figure.
7) Figures 4-15 must be larger for clear views. And please emphasize the best results in the figures, then conclude your contribution based on the reuslts.
8) You should add parameters for algorithms, and simulation time for each case. Then, please discuss if the simulation time is short enough to do a real work.
9) Please indicate the suit of a processor of computers for the simulation. What happy if the zone is very large about 100 km2 ? is it the simulation effective for the case. Please indicate all difficulties and advantages for the work.
10) Please give references for applied data. Is it the data large enough for the conclusion and contributions.
Comments on the Quality of English LanguageEnglish should be checked and revised through the paper. Typos must be checked too.
Author Response
Dear Editors and Reviewers:
Thank you for your letter and for the reviewers’ comments concerning our manuscript entitled “Design of an Intelligent Firefighting and Smart Escape Route Planning System Based on Improved Ant Colony Algorithm” (Manuscript Number: sensors-3201088). Those comments are all valuable and very helpful for revising and improving our paper, as well as the important guiding significance to our researches. We have studied comments carefully and have made correction which we hope meet with approval.
Revised portion are marked in yellow in the paper. The main corrections in the paper and the responds to the reviewers’ comments are as flowing:
Response to Reviewer #3:
[Comment 1] Only about 20 references were used for literature review. Please expand your overview by discussing at least 30 references.
Reply: We are very grateful to the reviewers for pointing out the shortcomings in our review section and the missing references.
How the paper is modified: We have added references, details are as follows:
1) In Introduction,supplemented the references1-4 concerning the factors influencing the spread of fire and added the abbreviation ACO for Ant Colony Optimization.
|
Before modification |
|
After modification |
2) In Literature Work,we have supplemented the references17-22 for the ant colony algorithm and its improved algorithms.
|
Before modification
|
|
After modification |
3)In Section 3.1.1, we have added reference 34 concerning the original Ant Colony Optimization algorithm.
|
Before modification
|
|
After modification
|
4) In section 3.1.2, references 35 and 36 and corresponding explanatory text for the Kriging interpolation algorithm have been added.
|
After modification
|
After supplementation, this paper cites a total of 37 references.
[Comment 2] The two first paragraphs in Introduction do not indicate the motivation of the study. Please point out studied objects, objective functions, and constraints. In addition, Please say why you choose the topic.
Reply: We are very grateful to the reviewers for pointing out the shortcomings in our research motivation, research subjects, and related details.
How the paper is modified: We make the following additions:
1)The introduction section has been rewritten, with the first paragraph explaining the motivation and reasons for researching this topic.
|
Before modification
|
|
After modification
|
2) In Section 3.1.3-1, formulas 9-12 and additional theoretical explanations were added to further detail the improvements in the evaluation and loss functions in the path planning algorithm after introducing the A* algorithm;
;
|
After modification |
3)in Section 3.1.3-2, formula 14 was added to explain the heuristic function of the improved algorithm, and formula 15 was added to explain the objective function of the algorithm in this paper:
|
Before modification
|
|
After modification |
[Comment 3] Algorithm 1 and Algorithm 2 must be named. And please use flowchart for them to clarify their applications.
Reply: Thank you to the reviewer for pointing out our oversights in the algorithm naming and flowchart representation.
How the paper is modified:Algorithm 1 is named 'IACO-A* algorithm', and Figure 2 is a supplementary flowchart of Algorithm 1; Algorithm 2 is named 'The fire detection and dynamic path planning algorithm', and Figure 3 is a supplementary flowchart of Algorithm 2.
|
Before modification |
|
After modification |
|
|
[Comment 4] The study lacks equations, only 6 equations for a study to be published in SCIE journal. This not satisfy the standard for a high-quality work as your paper.
Reply: We appreciate the reviewers pointing out the shortcomings in our theoretical explanations.
How the paper is modified:We make the following additions:
1)In section 3.1.2, equation 5-7 for the Kriging interpolation algorithm have been added.
|
After modification |
2)Section 3.1.3-1 includes additional details on formulas 9-12, providing a more comprehensive explanation of the improvements to the evaluation and loss functions in path planning algorithms after introducing the A* algorithm.
|
Before modification |
|
After modification |
3)In Section 3.1.3-2, adds Formula 14 to illustrate the heuristic function of the improved algorithm and adds Formula 15 to explain the objective function of the proposed algorithm.
|
After modification |
There are 15 formulas in the revised version of this document.
[Comment 5] In figure 3, you show many equations but you did not clarify them in body text. Please put them in body text and clarify them by explain their purpose, define each symbol and cite reference for each.
Reply: We appreciate the reviewers pointing out the shortcomings in our theoretical explanations.Please allow us to clarify:The original Figure 3 has been renumbered to Figure 4 after revision,Revised-Figure 4 shows the program of the improved ant colony algorithm written in MATLAB and called by LABVIEW, while Revised-Figure 2 presents the corresponding algorithm flowchart. Formulas 1-15 after supplementation are the respective formulas for the improved ant colony algorithm.
|
Before modification |
|
After modification |
[Comment 6] Check your typos and English: "Figure1, Figure2, Figure3", check the caption of figure.
Reply: We are very grateful to the reviewers for their reminders regarding writing and naming conventions.We have echecked and corrected the entire text's wording and English spelling, reviewed the descriptions of figures, tables, and formulas to ensure there were no omissions or errors.
[Comment 7] Figures 4-15 must be larger for clear views. And please emphasize the best results in the figures, then conclude your contribution based on the reuslts.
Reply: We greatly appreciate the reviewer pointing out our shortcomings in graphical representation.
How the paper is modified:The size of the images was rechecked and adjusted to ensure each one is clear. Figures 9 and 12 show the results of the path planning simulation, with annotations added in Figures 9 and 12 to indicate the shortest path for each scenario. In the final paragraph of section 4.2.2, a summary of the contributions of the algorithm in this paper was added based on the simulation results from Figures 9 and 12.
|
Before modification |
|
After modification |
[Comment 8] You should add parameters for algorithms, and simulation time for each case. Then, please discuss if the simulation time is short enough to do a real work.
Reply: Thank you for pointing out the insufficiency in our explanation regarding the detailed parameters and effectiveness of the algorithm.
How the paper is modified:We make the following additions:
1)In section 4.1, algorithm parameters were supplemented;
|
Before modification |
|
After modification |
2)In section 4.5, figure 16 was added to illustrate the convergence of different algorithms.
|
After modification |
3)In table 5, simulation time for each case was supplemented, and the issue of simulation time and algorithm real-time performance was discussed in the corresponding data analysis.
|
Before modification |
|
After modification |
[Comment 9] Please indicate the suit of a processor of computers for the simulation. What happy if the zone is very large about 100 km2 ? is it the simulation effective for the case. Please indicate all difficulties and advantages for the work.
Reply: We are very grateful to the reviewers for pointing out the insufficient explanation of the experimental operating environment in the paper, as well as the potential difficulties that the method may encounter in practical applications.
How the paper is modified: Section 4.1 explains the computer configuration used for the simulation.
Please allow me to clarify:As the area of the fire evacuation path planning expands, the complexity of the path increases, and the amount of computation does indeed grow, which is a challenge in this work. The consideration is to integrate this algorithm into the public fire system to leverage processor-level computing power for larger-scale fire monitoring and personnel evacuation guidance. These are the potential obstacles faced in this work. The advantage of this work lies in achieving synchronization between fire alarm monitoring and path planning, and it utilizes existing fire alarm monitoring equipment, keeping implementation costs low.
|
Before modification |
|
After modification |
[Comment 10] Please give references for applied data. Is it the data large enough for the conclusion and contributions.
Reply: We greatly appreciate the reviewers pointing out concerns regarding data reliability.
Data Availability Statement: The original contributions presented in the study are included in the article/supplementary material, further inquiries can be directed to the corresponding author/s.
According to the reviewers’ comments, we have made extensive modifications to our manuscript and supplemented extra data to make our results convincing. Thank you again for your positive comments and valuable suggestions to improve the quality of our manuscript.
We appreciate for editors and reviewers’ warm work earnestly, and hope that the correction will meet with approval.
Yours sincerely,
Song Yafei
September 18, 2024

Reviewer 4 Report
Comments and Suggestions for Authors
The paper introduces a route planning system based on improved ACO. The topic is interesting. The theoretical background is solid and the results are adequate. However, there are some minor points should be addressed. My suggestion is minor revision before acceptance. Here are some points the author should concern:
1. In the abstract, the author should state what is the main problem exists in the current studies and the method proposed in this paper can solve it. This is the key motivation of your research, it should be clear. Also, it would be helpful to state it in Introduction.
2. The limitations of the research should be addressed in Conclusion.
3. The authors should list their main contributions in the Introduction.
4. The literature review should be improved, some new references should be considered. For example, planning in marine domain (https://doi.org/10.1016/j.knosys.2024.112449)
Author Response
Dear Editors and Reviewers:
Thank you for your letter and for the reviewers’ comments concerning our manuscript entitled “Design of an Intelligent Firefighting and Smart Escape Route Planning System Based on Improved Ant Colony Algorithm” (Manuscript Number: sensors-3201088). Those comments are all valuable and very helpful for revising and improving our paper, as well as the important guiding significance to our researches. We have studied comments carefully and have made correction which we hope meet with approval.
Revised portion are marked in yellow in the paper. The main corrections in the paper and the responds to the reviewers’ comments are as flowing:
Response to Reviewer #4:
[Comment 1] In the abstract, the author should state what is the main problem exists in the current studies and the method proposed in this paper can solve it. This is the key motivation of your research, it should be clear. Also, it would be helpful to state it in Introduction.
Reply: We are very grateful to the reviewers for pointing out the shortcomings in the explanation of the research motivation in this paper.
How the paper is modified: We have each made the following additions
1) Further details about the research motivation of this paper were added in abstract .
|
Before modification
|
|
After modification
|
2) In Introduction, additional details were provided regarding the motivation for considering the fire scene situation in escape route planning
|
Before modification |
|
After modification |
3)In Literature Work, an analysis of the advantages and disadvantages of the ant colony algorithm was added, highlighting the motivation for proposing improvements to the ant colony algorithm.
|
After modification |
[Comment 2] The limitations of the research should be addressed in Conclusion.
Reply: We are very grateful to the reviewers for pointing out the shortcomings in our manuscript.
How the paper is modified: The conclusion section supplements the shortcomings of this article and proposes plans for future improvements.
|
Before modification |
|
After modification |
[Comment 3] The authors should list their main contributions in the Introduction.
Reply: We are very grateful to the reviewers for this suggestion.
How the paper is modified: .Author Contributions were added after the conclusion of the article and before the references.:
|
After modification |
[Comment 4]The literature review should be improved, some new references should be considered. For example, planning in marine domain (https://doi.org/10.1016/j.knosys.2024.112449)
Reply: We are very grateful to the reviewers for pointing out the deficiencies in the citation of literature in this paper.
How the paper is modified: We have each made the following additions:
1)In Introduction, supplemented with references regarding the spread of the fire.
|
Before modification |
|
After modification |
2)In Literature Work, added references on the ant colony algorithm and its improved versions.
|
After modification |
3)In section 3.1.2, references related to the Kriging interpolation algorithm have been added.
|
After modification |
According to the reviewers’ comments, we have made extensive modifications to our manuscript and supplemented extra data to make our results convincing. Thank you again for your positive comments and valuable suggestions to improve the quality of our manuscript.
We appreciate for editors and reviewers’ warm work earnestly, and hope that the correction will meet with approval.
Yours sincerely,
Song Yafei
September 18, 2024

Round 2
Reviewer 1 Report
Comments and Suggestions for Authors
The author can address all of my question. Now this paper is ready for publication.
Comments on the Quality of English LanguageMinor editing of English language required.
Reviewer 3 Report
Comments and Suggestions for Authors
The authors have addressed all my comments. So, the paper can be published in the journal.